# Self-Taught Recognizer: Toward Unsupervised Adaptation for Speech Foundation Models

Yuchen Hu[1,†]  Chen Chen[1,†]  Chao-Han Huck Yang[2]  Chengwei Qin[1]
Pin-Yu Chen[3]  Eng Siong Chng[1]  Chao Zhang[4]
[1]Nanyang Technological University  [2]NVIDIA Research
[3]IBM Research  [4]Tsinghua University
{yuchen005, chen1436}@e.ntu.edu.sg, hucky@nvidia.com

## Abstract

We propose an unsupervised adaptation framework, Self-TAught Recognizer (STAR), which leverages unlabeled data to enhance the robustness of automatic speech recognition (ASR) systems in diverse target domains, such as noise and accents. STAR is developed for prevalent speech foundation models based on Transformer-related architecture with auto-regressive decoding (e.g., Whisper, Canary; SeamlessM4T). Specifically, we propose a novel indicator that empirically integrates step-wise information during decoding to assess the token-level quality of pseudo labels **without** ground truth, thereby guiding model updates for effective unsupervised adaptation. Experimental results show that STAR achieves an average of 13.5% relative reduction in word error rate across 14 target domains, and it sometimes even approaches the upper-bound performance of supervised adaptation. Meanwhile, we observe that STAR prevents the adapted model from the catastrophic forgetting problem without recalling source-domain data. Furthermore, STAR exhibits high *data efficiency* that only requires less than one-hour unlabeled data, and seamless *generality* to alternative large speech models in recognition and translation tasks. Our code is publicly available at: https://github.com/YUCHEN005/STAR-Adapt.

## 1 Introduction

Human speech, characterized by its inherent acoustic nuances [70] and variability across speakers [27], is further complicated by the diverse and unpredictable environments. These factors contribute to significant domain distinctions in the speech signal, with differences in accent, speaking style, and background noise (visualized in Appendix B). Consequently, this diversity poses significant challenges in the field of automatic speech recognition (ASR), especially under diverse conditions [51].

In recent years, advancements in ASR technology [30, 84, 12, 73] have been boosted, primarily by the use of deep neural models and supervised learning with high-quality datasets. In particular, end-to-end ASR models pre-trained on industry-scale datasets have been made publicly available to the research community, such as OpenAI Whisper [73], Meta SeamlessM4T [4] and NVIDIA Canary [71]. Considering the high diversity of speech domains, even a well-trained ASR foundation model usually performs less satisfactorily when encountering a domain shift problem [49, 83, 85]. This performance degradation stems from a critical dilemma: collecting and labelling sufficient training data in the target domain is immensely time-consuming and labour-intensive, thus hindering the domain adaptation process of ASR models. Some existing efforts [32, 44] focus on leveraging labelled source domain and unlabeled target domain data to enhance the ASR performance, as shown

---

† Equal Contribution.

38th Conference on Neural Information Processing Systems (NeurIPS 2024).

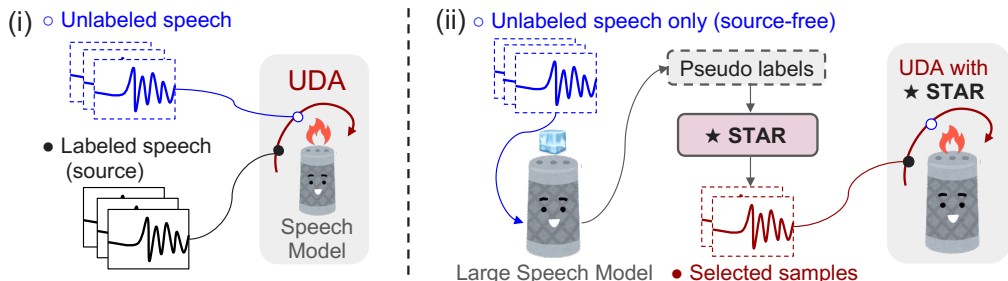

Figure 1: Illustration of unsupervised domain adaptation (UDA) and source-free UDA frameworks. (i) UDA problem. (ii) Source-free UDA by self-training. STAR works by selecting high-quality pseudo labels and guiding the ASR foundation model's adaptation at the token level.

in Fig. 1 (i). This solution is generally known as unsupervised domain adaptation (UDA) [24, 32, 44] and has been widely explored in both machine learning and speech processing communities.

In the context of the UDA problem in ASR, the human "self-directed" ability [39, 17] when encountering an unfamiliar speech domain is first illustrated. Despite the unawareness of the ground truth labels of our heard speech, individuals can learn speech-to-text mapping from their self-directed transcriptions, particularly when they have high confidence (see Fig. 8). This learning mechanism has a parallel in machine learning, known as "self-training" [75, 90, 41], which typically involves two stages. First, a pre-trained model generates the pseudo labels on target-domain data. Then, these data with pseudo labels, along with the associated confidence levels, are used to adapt the model.

Meanwhile, unlike approaches in existing ASR literature, which often require the source data (data used to pre-train the ASR model in source domains) to achieve UDA [63, 15, 5], humans, as the gold standard of speech communication, can address UDA issues in ASR without requiring any source data. Considering the exhibited generality of the speech foundation models with Transformer-related architectures based on the attention mechanism, it is the opportune moment to centre the attention mechanism on addressing the *source-free* UDA problem within the realm of ASR. Specifically, we study to adapt the pre-trained Whisper model using a small amount of unlabeled data from the target domain to become a domain-specific speech recognizer in different scenarios without using any source data, based on the process analogous to the human speech recognition as shown in Fig. 1 (ii). We hereby highlight the significant potential value that research on source-free UDA contributes to general ASR applications [3]: (i) It circumvents the extensive computational resources by adapting the ASR models without using any source data. (ii) It can considerably improve ASR performance in the target domain using only a small amount of speech samples without ground-truth labels.

In this work, we propose a source-free UDA approach called **S**elf-**TA**ught **R**ecognizer (STAR), which aims to enhance the performance of speech foundation models in specific target domains with unlabeled data. Based on the typical self-training scheme [92], STAR delves deeply into a general issue: Given the absence of ground-truth labels, how do we assess the quality of pseudo labels for guiding self-training? Unlike humans who can intuitively gauge their confidence in listening, the decoding "confidence scores" from attention-based ASR models are typically approximated by the pseudo posterior probabilities from softmax function [53] , which may be unreliable due to the well-known over-confident issue of softmax [37]. Traditionally with HMM-based ASR, the confidence scores can be estimated based on lattice and confusion network data structures [18, 61, 91], which, however, are difficult to obtain effectively in the end-to-end ASR framework.

In pursuit of a better quality indicator, we explore the self-attention matrix obtained during auto-regressive decoding, as it is not only grounded on speech input but also focuses on linguistic acceptability [36]. Specifically, we find the aggregated attention weights can be a more reliable indicator for measuring the quality of ASR-decoded tokens than the confidence scores. However, such an attentive score suffers from numerical instability, as recent findings [82, 62] from linguistic perspectives, it is normal for equally correct words (e.g., prepositions and nouns) to receive different semantic roles in a text sentence. This leads to the sub-optimality of using attentive scores alone to guide the fine-tuning process. We first substantiate these observations experimentally and then, in our STAR method, propose a novel integration approach based on their distinct characteristics, resulting in a both stable and reliable STAR indicator. Finally, it is employed to guide the subsequent finetuning process in a re-weighting manner, making a specific form of instructive adaptation.

Our experiments evaluate the proposed STAR in various practical scenarios, including background noise, speaker accents, and specific scenarios (e.g., interviews and talks). Comprehensive results show the significant gains from STAR that enhances Whisper by an average of **13.5%** relative word error rate (WER) reduction across 14 target domains. On some corpora, unsupervised STAR even approaches the upper bound of supervised adaptation using real labels. We also surprisingly observe that with informed finetuning, STAR prevents the adapted models from the common catastrophic forgetting problem without recalling source-domain data. Furthermore, we demonstrate that STAR enjoys: (i) *remarkable data efficiency:* it requires less than one hour of unlabeled data to adapt Whisper to its best performance on target domains; (ii) *seamless generality:* it is applicable to many prevalent speech foundation models and can be easily extended to the speech translation task.

In general, our contributions are summarized as follows:

- We direct our focus on source-free UDA in ASR with as one setting closed to real-world applications, where only a pre-trained speech foundation model and unlabeled speech samples are required to adapt to specific target domains.

- We present a score-based self-training approach called STAR that includes a novel indicator to evaluate the pseudo-label quality and achieve informed finetuning, which significantly enhances the domain-specific capabilities of speech foundation models across a wide range of target domains, including noise, accent, and specific scenarios.

- Intensive experiments demonstrate that STAR effectively avoids the common catastrophic forgetting problem in adaptation. Our further analysis of data efficiency and generality shows its potential for real-world applications, such as incremental updates for voice assistant.

## 2 Related Work

**Unsupervised Domain Adaptation in ASR.** Since acquiring the ground truth speech transcriptions is often prohibitively expensive in the target domain, many existing efforts bootstrap from available out-of-domain data to build an improved target domain model [79, 59, 93]. Besides directly simulating the target domain speech [31, 7], adversarial learning is frequently utilized to learn invariant representations to mitigate domain shifts [33, 23], which is also applied for front-end speech enhancement [64]. Meanwhile, teacher-student learning provides an alternative solution for efficient adaptation [50, 65]. These methods are also semi-supervised [86], since labels from the source domain are available. More recently, self-supervised pre-trained models (e.g. wav2vec2 [2]) have been used for pseudo-labelling to achieve unsupervised adaptation [44, 38].

**Source-free Unsupervised Domain Adaptation.** Given the potential presence of sensitive information in the source data [78], there is a high demand for source-free UDA methods that transfer a pre-trained source model to the unlabeled target domain without any source data [47, 66, 16]. As a long-discussed machine learning issue, the mainstream solutions include self-supervised knowledge distillation [57], contrastive learning [35], hidden structure mining [89], and uncertainty-guided adaptation [20]. Considering the inherent uncertainty in ASR decoding, we focus on the latter category and briefly review some representative indicators of uncertainty. Recently, there are some works [9] suggesting measuring uncertainty by the predicted variance from Monte Carlo Dropout [42], utilizing aleatoric uncertainty by encouraging intra-domain consistency [48], performing pseudo-labeling denoising using soft label correction [87], and introducing self-entropy descent mechanism to find a threshold for pseudo-labeling [54]. It is worth noting that, confidence estimation for ASR systems can be dated back for decades, starting by using lattices and confusion networks [18, 61] for HMM-based systems. Improved confidence estimation can be achieved by model-based approaches, such as conditional random fields [76], recurrent neural networks [40, 74] and graph neural networks [52]. More recent efforts [60, 77] focus on predicting uncertainty for auto-regressive decoding of attention-based models, however, they have not been applied in the source-free UDA.

**Summary.** Given the large amount of data used to pre-train the speech foundation models, it is difficult to define the scope of its source domain and keep the source data for re-training. Therefore we believe it is necessary to directly adapt speech foundation models to target domains for UDA for speech tasks. The proposed STAR method aims to assess the quality of pseudo labels produced by the auto-regressive decoding process, which leads to an instructive and effective self-training process. Since STAR can remove the need for keeping and retraining with source data and considerably reduce the performance difference between using ground truth and pseudo labels for adaptation with target

domain data samples, it has the potential to fulfil the goal of source-free UDA for the ASR task and achieve user-friendly deployment for real-world speech-based artificial intelligence products.

## 3 Methodology

### 3.1 Problem Setup

**ASR Formulation.** An end-to-end ASR system relies on a neural model $f$ to recognize the input speech $x \in \mathbb{R}^T$ into the corresponding text transcription $y \in \mathbb{R}^L$, where $T$ and $L$ denote the lengths of the input waveform and output text sequences respectively. During training, the model $f$ is optimized by teacher-forcing [46] with cross-entropy loss:

$$\mathcal{L}_{\text{ASR}}(x, y) = \sum_{l=1}^{L} -\log \mathcal{P}_\theta(y_l | y_{l-1}, \cdots, y_1, x), \tag{1}$$

where $y_{1:L}$ denotes the tokens in ground-truth labels $y$, and $\theta$ denotes the trainable parameters in $f$.

**UDA Setting.** Given a source ASR model $f^{(s)}$ trained on labelled source domain data $\{\mathcal{X}^{(s)}, \mathcal{Y}^{(s)}\} \in \mathcal{D}^{(s)}$, domain adaption in ASR aims to transfer the learned knowledge and obtain a model $f^{(t)}$ that performs well on target domain $\mathcal{D}^{(t)}$, i.e., $f^{(t)} : \mathcal{X}^{(t)} \rightarrow \mathcal{Y}^{(t)}$. UDA is required if ground-truth labels $\mathcal{Y}^{(t)}$ are not available. Source-free UDA [19, 55] posts a more challenging but practical scenario, where the source data $\{\mathcal{X}^{(s)}, \mathcal{Y}^{(s)}\}$ used to pre-train the ASR is no longer available in adaptation. That is, only speech inputs $\mathcal{X}^{(t)}$ is available when adapting the source model $f^{(s)}$ to the target domain $\mathcal{D}^{(t)}$. **Self-training Strategy.** In source-free UDA, since a source model itself typically generates pseudo-labels, some previous works [80] have referred to this learning approach as *semi-supervised learning*. To distinguish it from *unsupervised* domain adaptation, in this paper, we refer to the approach for addressing source-free UDA as *self-training*, consistent with the terminology used in studies [92]. Specifically, we adopt the pipeline of *pseudo-labeling* and *informed finetuning*. First, $N^{(t)}$ unlabeled speech segments $\mathcal{X}^{(t)} = \{x_i^{(t)}\}_{i=1}^{N^{(t)}}$ are fed into source model $f^{(s)}$ to generate the pseudo labels corresponding to each of them, which are denoted as $\hat{\mathcal{Y}}^{(t)} = \{\hat{y}_i^{(t)}\}_{i=1}^{N^{(t)}}$. Then, the paired dataset with the speech inputs and their newly-generated pseudo labels $\{\mathcal{X}^{(t)}, \hat{\mathcal{Y}}^{(t)}\}$ are used to finetune the source model to the target domain based on the self-training loss $\mathcal{L}_{\text{ST}}$:

$$\mathcal{L}_{\text{ST}}(\mathcal{X}^{(t)}, \hat{\mathcal{Y}}^{(t)}) = \sum_{i=1}^{N^{(t)}} \mathcal{L}_{\text{ASR}}(x_i^{(t)}, \hat{y}_i^{(t)}), \tag{2}$$

where the ASR loss $\mathcal{L}_{\text{ASR}}$ follows the definition in Eq. (1).

**Summary**. Since self-generated pseudo labels [63] do not introduce extra supervised information to the ASR source model, simply repeating this process is *unlikely* to yield performance improvements [75]. However, if high-quality pseudo labels are selected as domain-specific exemplars to inform the speech foundation model, it would then update in a direction beneficial to the target domain performance. Therefore, we propose a critical research question: How can we *assess the quality of pseudo labels* using an indicator that can also *guide the model's update*? The subsequent content of this section will delve into a detailed discussion from both token and utterance levels.

### 3.2 Token-level Assessment and Re-weighting

The auto-regressive decoding in ASR can provide step-wise information on predicted tokens, which can be used for token-level uncertainty assessment [60, 77]. More importantly, this information can guide the subsequent training process: assigning different weights to each token when calculating the CE loss in Eq.(2), namely *informed finetuning*.

**Why is *confidence* not a good indicator?** The confidence score denotes the highest value among the posterior probability predicted by a neural model. In auto-regressive decoding, the $l$-th step of token confidence score $C_l$ can be denoted as:

$$\mathcal{C}_l = \max \ \mathcal{P}(\hat{y}_l | \hat{y}_{l-1:1}, x, \theta^*). \tag{3}$$

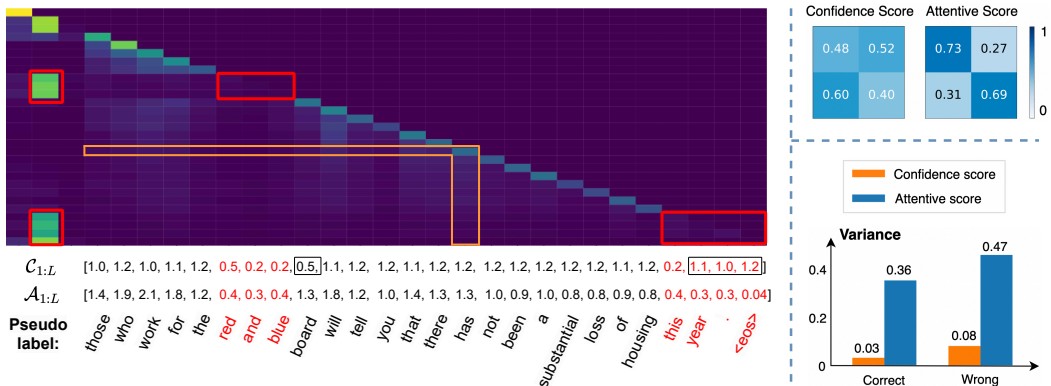

Figure 2: **(Left):** An example of pseudo label, ground-truth transcription, confidence scores, attention matrix and attentive scores. **(Right-Up):** Confusion matrix of confidence and attentive scores, where the y-axis denotes the pseudo token is correct or wrong, and the x-axis denotes the corresponding score is high or low (with 1 as the threshold, more analysis is in Fig. 6), so that the diagonal values indicate the score's reliability in assessing the quality of pseudo-label. **(Right-Down):** Variance of the two scores of correct and wrong pseudo tokens.

By preserving the $\mathcal{C}$ for each token during pseudo-labeling, we can perform informed finetuning with a re-weighting loss as follows:

$$\widetilde{\mathcal{L}}_{\text{ASR}}(x, \hat{y}) = \sum_{l=1}^{L} -\log \mathcal{P}_\theta(\hat{y}_l | \hat{y}_{l-1:1}, x) \cdot \mathcal{C}_l. \tag{4}$$

However, a substantial body of existing research [81] indicates that confidence does not accurately reflect predictive accuracy, especially in auto-regressive decoding [60]. In Eq. (4), the prediction of the current token is influenced by previously predicted tokens $\hat{y}_{l-1:1}$, which can easily lead to error accumulation and propagation. We further inspect this claim in Whisper by empirical observation. As shown in Fig. 2 (Right-Up), we employ a confusion matrix to visualize the relationship between confidence score and pseudo-label quality, which shows that 52% of correct tokens are assigned low confidence and 60% of wrong tokens are assigned high confidence (more discussion is in Appendix C). Therefore, confidence cannot be a reliable pseudo-label quality indicator alone, like discussed in [21].

**Is *attentive score* a better indicator?** We explore if the self-attention matrix $W$ obtained during auto-regressive decoding can reflect the pseudo-label quality. Unlike $\mathcal{C}_l$ defined in Eq. (3), $W$ has a direct association with $\mathcal{X}$ and linguistic acceptability [72], which means that it might be less influenced by the variability of speech input (see example in Fig. 2).

**Empirical Observation.** Starting from the fourth row and fourth column (first 3 tokens are fixed prompts: "$\langle|\text{en}|\rangle\langle|\text{transcribe}|\rangle\langle|\text{notimestamps}|\rangle$"), for the correctly decoded tokens (black), the attention weights are concentrated on the diagonal and partially fall on other pseudo tokens. However, for wrongly decoded tokens (red), the attention weights almost all fall on the second column that corresponds to the task prompt token "$\langle|\text{transcribe}|\rangle$" (highlighted in red boxes). To quantify this finding into a numerical metric, we defined an "aggregate pattern" indicator called *attentive score*, which is highlighted in the orange box in Fig. 2 and formulated as:

$$\mathcal{A}_l = \sum_{j=4}^{l} W_{l,j} + \sum_{i=l+1}^{L} W_{i,l}, \tag{5}$$

where $\mathcal{A}_l$ indicates the global semantic correlations between pseudo token $\hat{y}_l$ with all tokens $\{\hat{y}_l\}_{l=4}^{L}$ (first 3 tokens are task prompt). Specifically, we add the second term to also consider the attention weights with respect to future tokens, in order to capture the comprehensive global context to better assess the role of current token (see Table 7 for ablation study). We compare the values of attentive score $\mathcal{A}_l$ and confidence score $\mathcal{C}_l$ for this sentence in Fig. 2 (Left). As marked by black boxes, $\mathcal{C}_l$ provides unreliable assessments for both '*board*' (correct but low $\mathcal{C}_l$) and '*year . $\langle|\text{eos}|\rangle$*'(wrong but high $\mathcal{C}_l$). In comparison, $\mathcal{A}_l$ can accurately reflect the correctness of these tokens. To avoid randomness, we analyze CHiME-4 *test-real* and plot a confusion matrix in Fig. 2 (Right-Up). It is evident that, compared to $\mathcal{C}_l$, our $\mathcal{A}_l$ more reliably assesses the quality of predicted tokens.

Despite reliability, $\mathcal{A}_l$ exhibits less numerical stability, e.g., "for" and "housing" are both correct tokens but their $\mathcal{A}_l$ are distinct (1.8 vs. 0.8). The underlying reason is that their roles in the global context as prepositions and nouns are indeed different [82, 62]. However, when we try to use this $\mathcal{A}_l$ to guide the ASR loss re-weighting like Eq.(4), these labels are expected to be assigned comparable weights as they are equally correct. We verify this finding with the variance of $\mathcal{A}_l$ and $\mathcal{C}_l$ in Fig. 2 (Right-Down). For both correct and wrong tokens, $\mathcal{A}_l$ exhibits higher variance, indicating it may not be suitable to guide the finetuning in a re-weighting manner directly.

**STAR Indicator: Reliable and Stable.** To integrate the advantages of $\mathcal{C}_f$ and $\mathcal{A}_f$, we introduce a new indicator that balances reliability and stability. Specifically, in cases where $\mathcal{C}_f$ and $\mathcal{A}_f$ exhibit conflicting values toward a pseudo token, we would select $\mathcal{A}_f$ as an indicator that shows higher reliability. It can be mathematically formulated as:

$$\mathcal{S}_l^{\mathrm{conf}} = [\sigma(\mathcal{A}_l^2/\mathcal{C}_l - \lambda) + \sigma(\mathcal{C}_l^2/\mathcal{A}_l - \lambda)] * \mathcal{A}_l, \tag{6}$$

where $\sigma$ denotes the sigmoid function $\sigma(x) = 1/(1 + e^{-x})$, and here it simulates the step function to capture the cases of conflicting scores. Our definition of conflict is $\mathcal{A}_l^2/\mathcal{C}_l$ larger than a hyper-parameter threshold $\lambda$. This criterion can be decoupled into two terms, $\mathcal{A}_l$ and $\mathcal{A}_l/\mathcal{C}_l$, which means a large attentive score as well as a large gap between attentive and confidence scores[1]. Similarly, $\mathcal{C}_l^2/\mathcal{A}_l$ is another case of conflicting scores, and we add them up to simulate the logical "OR" operation.

On the other hand, if $\mathcal{A}_f$ and $\mathcal{C}_f$ present consistent assessment towards a pseudo token, $\mathcal{C}_f$ would be used to scale $\mathcal{A}_f$ using its stability. Specifically, we design a soft interpolation strategy inspired by focal loss [56] to integrate them:

$$\mathcal{S}_l^{\mathrm{cons}} = [\sigma(\lambda - \mathcal{A}_l^2/\mathcal{C}_l) * \sigma(\lambda - \mathcal{C}_l^2/\mathcal{A}_l)] * \mathcal{A}_l * e^{(\mathcal{C}_l - \mathcal{A}_l)/\tau}. \tag{7}$$

Similarly, we also use Sigmoid function to simulate the non-conflicting cases, where we multiply the two terms to denote logical "AND". Inspired by the smoothing technique in focal loss [56], we propose to leverage the gap between two scores for scaling $\mathcal{A}_l * e^{(\mathcal{C}_l - \mathcal{A}_l)/\tau}$, where $\tau$ is temperature.

During the subsequent *informed finetuning* stage, we combine the two indicators above to guide the training process in a re-weighting manner, and Eq.(4) should be re-written as:

$$\widetilde{\mathcal{L}}_{\mathrm{ASR}}(x, \hat{y}) = \sum_{l=1}^{L} -\log \mathcal{P}_\theta(\hat{y}_l|\hat{y}_{l-1:1}, x) * \mathcal{S}_l; \quad \text{where } \mathcal{S}_l = \mathcal{S}_l^{\mathrm{conf}} + \mathcal{S}_l^{\mathrm{cons}}. \tag{8}$$

As a result, the STAR scores are both reliable and stable as shown in Fig. 5, which serves as a better quality indicator to guide the *informed finetuning* (see Algorithm 1 in Appendix for details).

### 3.3 Utterance-level Filtering

The utterance-level filtering aims to remove those predicted utterances with low overall quality since they are probably harmful for subsequent adaptation. We now introduce several existing approaches to assess the utterance-level quality of pseudo labels, which are often used for uncertainty estimation. Notably, high uncertainty usually implicates low quality for the generated sequence.

**Monte Carlo Sampling** [42] conduct multiple times of stochastic forward decoding with *activated dropout* to get a list of predictions [9]. Then the list with a large variance is considered to have high uncertainty and should be removed from subsequent training. However, this method does not apply to Whisper as it does not use dropout in training. As an alternative, we introduce a similar method for assessing utterance-level uncertainty. Specifically, given an input speech $x$, we first implement one forward decoding and set the result $\hat{y}$ as the base transcription. Then, we randomly disturb the model weights of Whisper with Gaussian noise, and repeat the forward decoding for $K$ times, resulting in a list of pseudo transcriptions $\{\hat{y}_k\}_{k=1}^K$. Thereafter, we calculate the edit distance (ED) between pseudo transcription $\hat{y}_k$ and the base transcription $\hat{y}$, which indicates the impact of disturbance on Whisper decoding. Then, the model's robustness in transcribing speech $x$ can be calculated as:

$$U(x, \hat{y}) = \frac{1}{K} \sum_{k=1}^{K} ED(\hat{y}, \hat{y}_k). \tag{9}$$

---

[1]To avoid special cases like two tiny scores where one is many times of another (e.g., 0.01, 0.001).

Table 1: Main WER (%) results of the proposed STAR adaptation and baselines in various ASR domains. "Whisper (frozen)" denotes the zero-shot performance without adaptation. "Whisper (self-train.)" is the vanilla self-training scheme consisting of pseudo-labeling and finetuning. Based on that, "UTT$_{\text{filter}}$" adds utterance-level filtering explained in §3.3, and "TOK$_{\text{reweight}}$" performs two token-level re-weighting explained in §3.2. "Whisper (real label)" is supervised learning with real (ground truth) labels and can be viewed as the upper-bound performance of source-free UDA.

| Testing Scenario | | Whisper (frozen) | Whisper (self-train.) | UTT$_{\text{filter}}$ | TOK$_{\text{reweight}}$ $\mathcal{C}_l$ | $\mathcal{A}_l$ | **STAR (ours)** | Whisper (real label) |
|---|---|---|---|---|---|---|---|---|
| *Background Noise* | | | | | | | | |
| CHiME-4 | *test-real* | 6.8 | 6.9 | 6.4 | 6.5 | 6.2 | **6.0**$_{-11.8\%}$ | 5.2 |
| | *test-simu* | 9.9 | 10.1 | 9.7 | 9.8 | 9.5 | **9.4**$_{-5.1\%}$ | 8.7 |
| | *dev-real* | 4.6 | 4.5 | 4.3 | 4.3 | 4.1 | **3.9**$_{-15.2\%}$ | 3.2 |
| | *dev-simu* | 7.0 | 7.0 | 6.6 | 6.7 | 6.6 | **6.4**$_{-8.6\%}$ | 5.9 |
| LS-FreeSound | *babble* | 40.2 | 37.6 | 35.0 | 33.5 | 31.3 | **30.2**$_{-24.9\%}$ | 27.2 |
| | *airport* | 15.6 | 15.5 | 15.2 | 15.3 | 15.0 | **14.8**$_{-5.1\%}$ | 14.5 |
| | *car* | 2.9 | 3.0 | 2.8 | 2.8 | 2.6 | **2.5**$_{-13.8\%}$ | 2.4 |
| RATS | *radio* | 46.9 | 47.2 | 46.0 | 45.5 | 44.9 | **44.6**$_{-4.9\%}$ | 38.6 |
| *Speaker Accents* | | | | | | | | |
| CommonVoice | *African* | 6.0 | 5.8 | 5.5 | 5.4 | 5.0 | **4.8**$_{-20.0\%}$ | 4.6 |
| | *Australian* | 5.8 | 5.7 | 5.6 | 5.5 | 5.2 | **5.1**$_{-12.1\%}$ | 4.3 |
| | *Indian* | 6.6 | 6.5 | 6.3 | 6.4 | 6.1 | **6.0**$_{-9.1\%}$ | 5.7 |
| | *Singaporean* | 6.5 | 6.2 | 5.8 | 5.8 | 5.4 | **5.1**$_{-21.5\%}$ | 4.9 |
| *Specific Scenarios* | | | | | | | | |
| TED-LIUM 3 | *TED talks* | 5.2 | 4.9 | 4.7 | 4.8 | 4.3 | **4.1**$_{-21.2\%}$ | 3.6 |
| SwitchBoard | *telephone* | 13.3 | 13.0 | 12.7 | 12.3 | 11.9 | **11.7**$_{-12.0\%}$ | 9.9 |
| LRS2 | *BBC talks* | 8.5 | 8.3 | 7.6 | 7.9 | 7.4 | **7.0**$_{-17.6\%}$ | 5.6 |
| ATIS | *airline info.* | 3.6 | 3.5 | 3.3 | 3.3 | 3.2 | **2.9**$_{-19.4\%}$ | 2.0 |
| CORAAL | *interview* | 21.5 | 21.3 | 20.8 | 20.7 | 20.4 | **20.1**$_{-6.5\%}$ | 17.9 |

Table 2: WER (%) results regarding catastrophic forgetting. "Frozen" denotes Whisper zero-shot without adaptation. "Self-train" denotes the self-training baseline. "STAR" model is adapted to **CHiME-4** using STAR; then evaluated on other domains. More results are in Table 11.

| Model | LS-FreeSound *babble* | *airport* | *car* | RATS | CommonVoice *af* | *au* | *in* | *sg* | TED-3 | SWBD | ATIS |
|---|---|---|---|---|---|---|---|---|---|---|---|
| Frozen | 40.2 | **15.6** | 2.9 | 46.9 | **6.0** | **5.8** | **6.6** | 6.5 | 5.2 | **13.3** | 3.6 |
| Self-train. | 38.2 | 16.6 | 2.9 | 47.3 | 6.4 | 5.9 | 6.7 | 6.3 | 5.3 | 13.7 | 3.4 |
| STAR | **33.3** | 15.7 | **2.8** | **46.1** | 6.1 | **5.8** | 6.7 | **5.6** | **5.0** | 13.5 | **2.9** |

After obtaining $k$ pseudo labels, we can examine their diversity to further assess the model's uncertainty. If there are many repetitions in the list, it indicates that the model is more confident in transcribing speech $x$. Therefore, we utilize a scaling factor $l$ that is equal to the utterance amount after de-duplication. The final utterance-level quality is combined by the numeric multiplication of $l$ and $U(x, \hat{y})$, which is then used to rank the $N_t$ pseudo data samples, and top $\alpha\%$ samples are removed due to large data uncertainty. Additionally, we also implement a **beam search decoding** and a **consensus decoding** [61] baselines as alternative utterance-level filtering approaches for comparison, where more experimental results and discussions are presented in Appendix D.

## 4 Experimental Setup

### 4.1 ASR Domains

We introduce STAR in various ASR domains to verify its general effectiveness, including noisy speech, accented speech, and specific scenarios. First, for noisy speech we use the CHiME-4 [83], LibriSpeech-FreeSound [69], and RATS [26] datasets, which covers a wide range of noise types including bus, cafe, pedestrian area, street junctions, babble, car, airport, and the challenging radio communication noises. Second, we select four typical accents from the CommonVoice [1] dataset,

Table 3: Case study of an accented speech in CV-*in* (ID: "en_19795319"). The wrong tokens are highlighted in red. Variance indicates the stability of different scores. "NCE" denotes normalized cross-entropy, where a higher value indicates better measure quality (more results are in Fig. 5).

| Metric | Content | Variance | NCE Score |
|---|---|---|---|
| Ground-truth | they are organised by scientific themes. | - | - |
| Pseudo label | they are organised by scientific teams. | - | - |
| $\mathcal{C}_{1:L}$ | $[0.81, 0.88, 0.98, 1.21, 1.13, 1.17, 0.82]$ | 0.023 | $-0.671$ |
| $\mathcal{A}_{1:L}$ | $[1.47, 1.49, 0.95, 1.20, 0.79, 0.43, 0.67]$ | 0.101 | 0.146 |
| $\mathcal{S}_{1:L}$ (ours) | $[1.39, 1.40, 0.91, 1.14, 1.03, 0.41, 0.73]$ | 0.058 | 0.322 |

Table 4: WER (%) results of STAR with different speech foundation models on CHiME-4 *test-real*. More models / datasets are evaluated in Table 9 and 6.

| Model | Baseline | Self-train. | STAR | Real |
|---|---|---|---|---|
| Whisper-V3-1.5B | 6.8 | 6.9 | $6.0_{-11.8\%}$ | 5.2 |
| Whisper-Med-0.8B | 8.9 | 8.8 | $8.0_{-10.1\%}$ | 7.1 |
| OWSM-V3.1-1.0B | 8.4 | 8.1 | $7.5_{-10.7\%}$ | 6.5 |
| Canary-1.0B | 8.2 | 8.0 | $7.2_{-12.2\%}$ | 6.4 |
| Parakeet-TDT-1.1B | 8.0 | 7.8 | $7.0_{-12.5\%}$ | 6.2 |

Table 5: BLEU results of STAR on speech translation task with FLEURS [14] test sets.

| $X \rightarrow En$ | Baseline | Self-train. | STAR | Real |
|---|---|---|---|---|
| Ar | 21.9 | 22.1 | $23.3_{+1.4}$ | 24.5 |
| De | 33.7 | 34.0 | $35.9_{+2.2}$ | 36.5 |
| Es | 23.9 | 24.1 | $24.8_{+0.9}$ | 26.4 |
| Fa | 16.6 | 16.3 | $17.6_{+1.0}$ | 19.0 |
| Hi | 22.4 | 22.5 | $23.4_{+1.0}$ | 24.4 |
| Zh | 16.3 | 16.3 | $17.1_{+0.8}$ | 17.9 |

including African, Australian, Indian, and Singaporean accents. Finally, we also evaluate our approach under some specific scenarios, including BBC talks (LRS2 [13]), TED talks (TED-LIUM 3 [29]), telephone conversation (SwitchBoard [25]), interview conversation (CORAAL [43]), and airline information consultation (ATIS [28]). More details about the datasets are presented in Appendix F.

## 4.2 Configurations

We use the Whisper-Large-V3 model for main experiments, which contains 1.5 billion parameters trained on 680k-hour web-scale data. It is fine-tuned using Adam optimizer [45] with an initial learning rate of $1e^{-5}$ for 2 epochs. The batch size is set to 1 with 16 gradient accumulation steps. For hyper-parameters, the threshold $\lambda$ is set to 2 and the temperature $\tau$ is 10. In addition, the percentile $\alpha$ of utterance-level filtering is 20, which shows consistent effectiveness across different datasets.

## 5 Results and Analysis

### 5.1 Effectiveness of STAR

To examine the effectiveness of STAR, we conduct comparative experiments across various domains and report the WER results in Table 1.

**Main Results.** From noise adaptation results on CHiME-4, LS-FreeSound, and RATS, we observe that: (i) STAR enhances Whisper in all noise scenarios, reducing the WER up to 24.9% relatively. Specifically, on the challenging RATS dataset with pseudo labels of a 46.9% WER, our STAR can still produce a 4.9% relative improvement. (ii) For some domains, e.g., "*airport*" and "*car*", STAR can even approach the upper-bound performance by supervised learning. This demonstrates that even with unlabeled data only, our method can effectively adapt Whisper to specific target domains. From results on other domains, we observe that: (i) STAR consistently improves the accented ASR to approach the supervised upper bound. (ii) Whisper does not perform well in some colloquial scenarios (*SwitchBoard* and *CORAAL*) as the spoken language tends to be informal and less grammatically correct, which leads to poor-quality pseudo labels and then influences our adaptation performance.

**Analysis of Catastrophic Forgetting.** Table 2 analyzes the potential forgetting issue of our method by evaluating the *CHiME-4*-finetuned model on other datasets. Surprisingly, contrary to the common catastrophic forgetting issue that commonly happens in traditional source-free ASR adaptation, our STAR approach can even improve the performance in other domains. We speculate that under the self-training scheme, the pseudo label is generated by the model itself, so that it may avoid the model from over-fitting to samples with vastly different data distributions [10]. Furthermore, compared to vanilla self-training, STAR can better highlight the high-quality pseudo tokens for *informed finetuning*, which may help improve the model's general ASR ability. More detailed analysis are in §E.

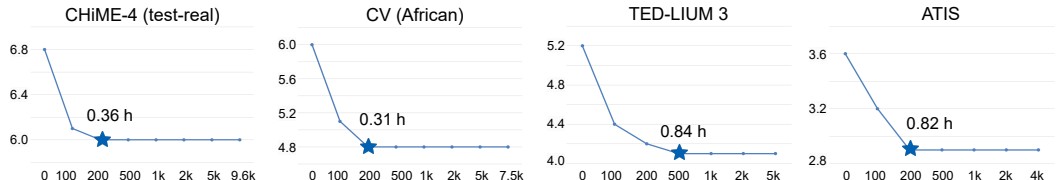

Figure 3: WER (%) results with different numbers of unlabeled training samples. The minimum required data amount (in hours) to obtain the best performance is highlighted in the star mark.

**Analysis of Indicators.** Table 1 also presents the performance of different indicators in the *informed finetuning*. First, we observe that the utterance-level filtering yields some effects by removing bad training samples. Then, for token-level re-weighting, the two pseudo-label quality indicators both improve the performance, where the attentive score performs better due to higher reliability. Our proposed STAR indicator achieves the best result by integrating the strengths of both scores. We also use a case in Table 3 to illustrate, where exists a wrong pseudo token "teams". The confidence score fails to reflect this error while the attentive score succeeds, but the latter suffers from less numerical stability (i.e., large variance). By integrating their strengths, our STAR score achieves both reliability and stability in assessing the pseudo-label's quality. In addition, we also calculate the NCE metric to show the better quality of our proposed STAR and attentive scores than traditional confidence scores.

## 5.2 Generality of STAR

**Generalization to Different Speech Foundation Models.** To further evaluate the generalization ability of our approach, we extend STAR to different foundation models, including OWSM, Canary and Parakeet-TDT that take top places in the HuggingFace ASR leader-board [2]. Consistent performance gains (i.e., over 10% relative WER reduction) on these models has verified the excellent generality of STAR. In addition, it also works well on relatively small models like Whisper-Medium.en-0.8B.

**Generalization to Speech Translation (ST) Task.** Apart from ASR, we also investigate another widely studied speech task, the ST task, to further verify the generality of STAR adaptation. As shown in Table 5, results on various FLEURS X→En tracks illustrate an average of over 1.2 BLEU improvements (2.2 BLEU for De→En). It shows the good potential of our STAR adaptation on other sequence-to-sequence tasks besides ASR, which could lead to more extensions for future work.

## 5.3 Ablation Study

In this section, we conduct ablation studies to analyze STAR from perspectives of data (Fig. 3), model (Table 9), and finetuning approaches (Table 10), which provide a constructive reference for deploying ASR foundation models in practical scenarios using STAR. More analysis are in Appendix E.

**Data Efficiency.** We explore the requirement of unlabeled data amount ($N_{t'}$) for STAR adaptation. Fig. 3 shows the WER results on four datasets with different numbers of training utterances. Surprisingly, only 200 to 500 sentences (less than 1-hour unlabeled speech data) are required to achieve the optimal effects, which cost around 0.8-hour training time on single NVIDIA-A100-40GB GPU. This remarkable data efficiency significantly saves the labours in real-world applications: not only is there no need for manual labelling, but the collection of unlabeled data also requires less than one hour.

**Model Size.** Table 9 reports the performance on CHiME-4 *test-real* that applies STAR to the Whisper family with different model sizes. Results show that our STAR adaptation works well on difference scales of foundation models. Specifically, the promising performance gains on light model (base.en) implicates the potential of STAR in practical resource-constrained conditions, such as mobile devices.

**Finetuning Approach.** Considering that adapting speech foundation models with a small amount of data might risk over-fitting, we explore the impact of different finetuning approaches in Table 10. We observe that both regular finetuning (full, encoder-only, decoder-only) and efficient finetuning methods (LoRA) yield similar effectiveness, which provides flexible choices under different settings.

---

[2]`https://huggingface.co/spaces/hf-audio/open_asr_leaderboard`

# 6 Conclusion

We propose STAR, a source-free UDA method that effectively adapts the speech foundation models to various target domains with unlabeled data. Specifically, STAR introduces a novel indicator to assess the pseudo-label quality and then instructively guide the finetuning of the model. Our experiments verify STAR's efficacy on ASR tasks across a wide range of target domains including noise, accent, and specific scenarios, and it even approaches the upper-bound performance of supervised adaptation on some corpora. Furthermore, we observe that STAR can avoid the catastrophic forgetting problem that is often suffered by models adapted without recalling source-domain data. Furthermore, STAR only requires less than one hour of unlabeled data to achieve an average of 13.5% relative WER reduction across 14 domains, and it also shows seamless generality to speech translation tasks. This enables us to deploy speech systems in real-world scenarios rapidly and conveniently.

## Acknowledgement

This research is supported by the National Research Foundation, Singapore, under its AI Singapore Programme grant number AISG2-100E-2022-102. Any opinions, findings and conclusions or recommendations expressed in this material are those of the author(s) and do not reflect the views of National Research Foundation, Singapore. The computational work for this article was partially performed on resources of the National Supercomputing Centre, Singapore (https://www.nscc.sg).

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

## A    Additional Discussions on the Design of the STAR Framework

*Question 1: Is the proposed STAR method limited to the Whisper model only?*

STAR is a general source-free UDA method that can be compatible with any attention-based speech foundation model. To validate this, we also use several other models in our experiments, including

OWSM-V3.1-1.0B [68][3], Canary-1.0B[4], Parakeet-TDT-1.1B[5], and SeamlessM4T-V2-2.3B [4][6]. Table 4 verify the effective generalization ability of our STAR adaptation method on various state-of-the-art ASR foundation models [7]. Furthermore, we also employ the speech translation foundation model SeamlessM4T to verify our generality. Table 6 presents the WER results on CHiME-4 test sets with SeamlessM4T, where the STAR adaptation achieves significant improvements over zero-shot and self-training baselines, which even approaches the supervised upper bound. We observed that although the performance of SeamlessM4T-Large-V2 is slightly worse than Whisper Large-V3 on ASR task, STAR can still achieve up to a 30.7% WER reduction that improves its noise robustness.

Table 6: WER (%) results of STAR with latest speech foundation model, SeamlessM4T-Large-V2 [4], on CHiME-4 test sets.

| Test Set | Baseline | Self-train. | STAR (ours) | Real label |
|---|---|---|---|---|
| test-real | 12.3 | 11.5 | $9.1_{-26.0\%}$ | 8.7 |
| test-simu | 15.2 | 15.0 | $13.2_{-13.2\%}$ | 13.0 |
| dev-real | 8.8 | 8.3 | $6.1_{-30.7\%}$ | 5.8 |
| dev-simu | 11.4 | 11.0 | $9.2_{-19.3\%}$ | 9.0 |

***Question 2****: Can STAR be applied to smaller or streaming ASR models like WavLM, RNN-T?*

STAR requires large speech models to possess *universal* robustness across various domains to fulfill their role as a reliable pseudo-labeler, where domain-specific ASR models like WavLM and RNN-T may not work well. To illustrate, the WavLM-Conformer ASR baseline [11] achieves the state-of-the-art result on the LibriSpeech test-clean dataset (with a WER of only 1.8%). However, when facing domain shifts, such as the CHiME-4 noisy dataset, its zero-shot performance drops to 14.4%, and it exceeds 20% on the CommonVoice accented dataset. Under these circumstances, it is nearly impossible to use only unlabeled data to adapt this model to the Whisper's zero-shot performance (5 7% WER). Therefore, we argue that adding such baselines is not of much reference value. As more general-purpose speech foundation models are released, we prefer to focus our research on studying their decoding behaviors to adapt them more efficiently and conveniently to specific task domains. Table 4 and 6 provide more results on CHiME-4 dataset with more speech foundation models (i.e., OWSM, Canary, Parakeet, SeamlessM4T) to show the good generality of STAR.

***Question 3****: Is the proposed STAR method limited to the ASR task?*

Our proposed STAR approach is compatible with any tasks using *attention-based encoder-decoder architecture with auto-regressive decoding*, the most prevalent framework in many areas not limited to speech and language. Therefore, we believe this work provides useful insights to researchers from other communities who use similar model architectures and need to assess the quality of auto-regressive decoders, such as speech translation, audio/image captioning. Table 5 presents the strong results of STAR on speech translation task, which verifies its task generality. However, since this work focuses on ASR task, we would like to leave the evaluation on more tasks to future work. In addition, recent research on LLMs [36] also focuses on the unsmooth self-attention matrix like Fig. 2, where they successfully alleviate the LLMs' hallucination problem in auto-regressive decoding through this observation. This evidence indicates the potential impact of our method on other communities.

***Question 4****: What is the difference between STAR with the existing self-training method in ASR?*

We summarize the vital difference in the following two points:

- Prior works focus on adapting from one source domain to a single target domain (e.g., clean to noisy [58]), whereas STAR leverages the universality of Whisper to explore one-to-many domain adaptation. Although the domain mismatch issue in the latter approach is less severe than in the former, we argue that the baseline performance of the latter is significantly better than that of the former, making improvements more challenging to achieve.

---

[3]`https://huggingface.co/espnet/owsm_v3.1_ebf`
[4]`https://huggingface.co/nvidia/canary-1b`
[5]`https://huggingface.co/nvidia/parakeet-tdt-1.1b`
[6]`https://huggingface.co/facebook/seamless-m4t-v2-large`
[7]`https://huggingface.co/spaces/hf-audio/open_asr_leaderboard`

- Although both of them are auto-regressive processes, the decoding of speech foundation models exhibits partially distinct characteristics compared with vanilla ASR decoders in previous works, such as over-confidence phenomena [37]. Therefore, STAR adopts an empirical indicator of quality derived from the decoding features of speech foundation models, which can more effectively guide the subsequent finetuning process.

***Question 5****: What is the scope of efficacy when applying STAR adaptation?*

As shown in Table 1, STAR can improve the performance with zero-shot WER ranging from 2.9% (car) to 46.9% (radio). However, we observe that the WER improvement in RATS mainly stems from the adjustment of some prepositions and articles, which may not improve the comprehensibility of recognition results fundamentally. Therefore, collecting labeled data for supervised learning is inevitable when the scenarios are quite challenging.

***Question 6****: Can STAR be involved in adapting test data?*

We observe that some works perform unsupervised domain adaptation using unlabeled test data. However, STAR only accesses test data once, and the unlabeled target domain data is drawn from the training set of corresponding corpus. We argue this setting more closely aligns with practical scenarios: developers cannot access the test set during the adaptation process. However, they can conveniently collect small amount (see data efficiency in Fig. 3) of unlabeled target-domain speech for adaptation, and then deploy the adapted model in testing environments.

***Question 7****: What about the broader impacts of this work?*

This work supports rapid and convenient deployment of ASR applications in real-world scenarios, which poses positive societal impact. During training process, we only use publicly available data and pre-trained models, so that our work will not pose explicit negative impact. One thing worth noting is that, our algorithm shows good generality and thus might cause abuse in some special occasions (e.g., confidential), we will release code carefully under strict licenses and rules to avoid negative impact.

***Question 8****: What about the limitations of this work?*

Our approach is designed specifically for Transformer-based speech foundation models, so that it may not handle the cases beyond the foundation models, e.g., the unseen languages, unseen tasks, extremely adverse conditions, etc.

## B    Visualization of Speech Domains Distinction

Fig. 4 visualizes the spectrograms of parallel clean and noisy speech samples. We can observe clear speech patterns in the clean spectrogram (i), while they are significantly contaminated by noise in the noisy spectrograms (ii) and (iii). Specifically, the babble noise corrupts speech signals more than airport noise, where speech patterns are almost completely removed. The reason is babble noise contains human speech and thus of sample type as an original speech signal, resulting in more significant corruption. These two kinds of noisy speech are reported in Table 1. Overall, the domains of clean and noisy speech are quite distinct from the perspective of pattern recognition.

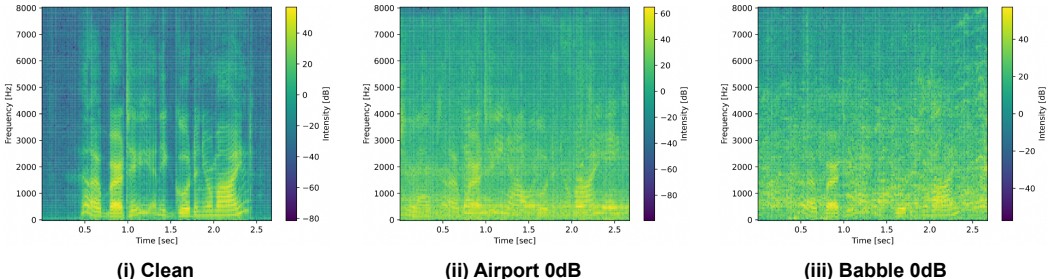

Figure 4: Spectrograms of parallel clean and noisy speech samples, where we select two noise types for visualization, i.e., airport station and babble (used in our experiments). The speech samples are selected from the LS-FreeSound test set, and the sample ID is "1089-134686-0003".

## C    More Discussions of Pseudo-label Quality Indicators

Fig. 5 presents more investigations of the quality indicators. First, from the confusion matrix we can observe the higher reliability of the attentive score over the confidence score, which has been discussed in Section 5.1, and our proposed STAR score maintains the high reliability of the attentive score by sophisticated integration. Then, from the variance statistics we can observe that the attentive score suffers from less numerical stability, while our proposed STAR score benefits from the high stability of the confidence score. As a result, the STAR score provides a both reliable and stable indicator of the quality of the pseudo label. Furthermore, we note that after STAR adaptation, the Whisper model can produce higher-quality pseudo labels, which not only verifies its effectiveness but also explains its potential for iterative adaptation (see Section E).

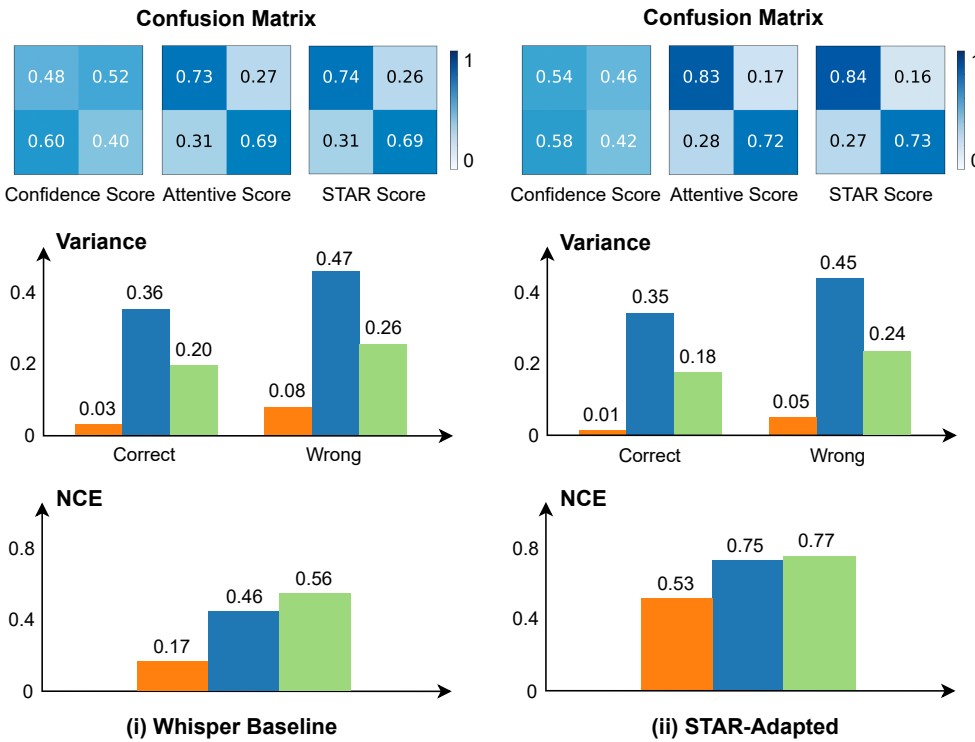

Figure 5: Confusion matrix, variance and normalized cross-entropy (NCE) of confidence score (**orange**), attentive score (**blue**), and our STAR score (**green**), in terms of the Whisper baseline and our STAR-adapted model. For the confusion matrix, the y-axis denotes the correctness of pseudo tokens, i.e., correct and wrong, and the x-axis denotes whether the corresponding score is high or low. Since all scores are normalized via divided by mean value, we set 1 as the threshold to separate them into large and small groups, where more thresholds are analyzed in Fig. 6. NCE is a statistical metric to measure the quality of confidence measure, where higher value indicates better measurement.

## D    More Discussion on Utterance-level filtering

**Beam search decoding** is a widely used strategy in sequence-to-sequence decoding, which expands the search space to obtain an N-best hypotheses list. Recent research on speech foundation models shows that N-best results contain rich information [8, 88], where the diversity can reflect the final WER performance [34, 6]. Specifically, we utilize the beam search decoding as a replacement for Gaussian disturbance and obtain an N-best hypotheses list for diversity calculation. The best hypotheses and beam size $N$ are respectively viewed as base transcription $\hat{y}$ and $K$, and the quality of utterance $U$ is calculated in the same manner as Eq.(9). Furthermore, we also introduce an earlier method from conventional ASR, consensus decoding [61], as an alternative to investigate the role of

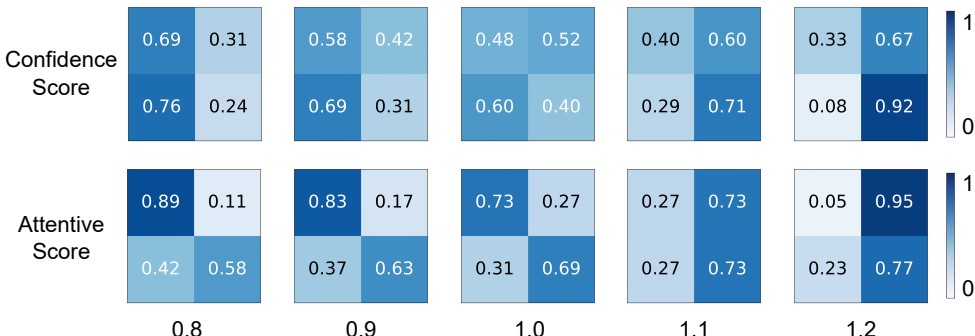

Figure 6: Confusion matrix of confidence score and attentive score in terms of different thresholds for separating large and small groups.

Table 7: Ablation study on employing different pseudo tokens to calculate the attentive score in Eq. 5 using CHiME-4 *test-real* data.

| Metric | History tokens | Future tokens | Both |
|---|---|---|---|
| NCE | 0.42 | 0.37 | 0.46 |
| WER (%) | 6.4 | 6.5 | 6.2 |

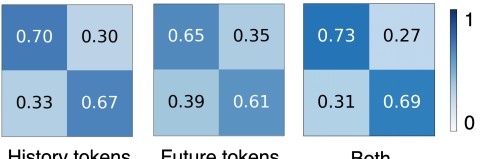

Figure 7: Confusion matrix of attentive score calculated using different pseudo tokens.

utterance-level filtering approach, where the utterance quality is estimated similar to the beam search decoding.

Table 8 presents the ablation study of utterance-level filtering strategy. First, we compare the $K$-hypotheses by Gaussian disturbance with the $N$-best hypotheses by beam search decoding and consensus decoding, where we observe that the former performs a little better than the latter two. Therefore, we employ Gaussian disturbance for utterance-level filtering in the main experiments. Finally, we investigate the role of utterance-level filtering in the entire STAR adaptation, where the results demonstrate its contribution in the final performance gains (second last column).

Table 8: Ablation study of utterance-level filtering in terms of Gaussian disturbance, beam search decoding, and consensus decoding [61].

| Testing Scenario | | Whisper (frozen) | Whisper (self-training) | UTT$_{filter}$ | | | STAR (ours) | | Real label |
|---|---|---|---|---|---|---|---|---|---|
| | | | | Gaussian | Beam | Consensus | w/o UTT | w/ UTT | |
| CHiME-4 | *test-real* | 6.8 | 6.9 | 6.4 | 6.6 | 6.6 | 6.2 | **6.0**$_{-11.8\%}$ | 5.2 |
| | *test-simu* | 9.9 | 10.1 | 9.7 | 9.8 | 9.7 | 9.7 | **9.4**$_{-5.1\%}$ | 8.7 |
| | *dev-real* | 4.6 | 4.5 | 4.3 | 4.3 | 4.4 | 4.0 | **3.9**$_{-15.2\%}$ | 3.2 |
| | *dev-simu* | 7.0 | 7.0 | 6.6 | 6.7 | 6.7 | 6.6 | **6.4**$_{-8.6\%}$ | 5.9 |

# E   Additional Ablation Study

Following Section 5.3, here we would like to present more details about the ablation study on model size and finetuning approaches. In addition, we also observe that our STAR method can further improve the performance with multiple-round iterative adaptation.

**Model Size.** Table 9 reports the performance on CHiME-4 *test-real* that applies STAR to the Whisper family with different model sizes. We can observe consistent and significant improvements on different scales of Whisper models. Specifically, although the light model (base.en) exhibits poor noise robustness, our STAR can bring 45.4% relative improvement. It indicates the potential value of STAR adaptation in practical resource-constrained conditions, such as mobile devices.

**Finetuning Approach.** Considering that training large speech models with a small amount of data might risk over-fitting, we explore the impact of different tuning approaches for the informed finetuning process in this experiment. From Table 10, we observe that (i) freezing part of parameters can not improve the performance of STAR adaptation. However, finetuning the decoder only is the

Table 9: WER (%) results of different model sizes on CHiME-4 *test-real* set. "# Param." is the number of model parameters. "Real" denotes Whisper with real label finetuning.

| Model Size | # Param. | Baseline | STAR | Real |
|---|---|---|---|---|
| large-v3 | | 6.8 | $6.0_{-11.8\%}$ | 5.2 |
| large-v2 | 1,550 M | 7.7 | $6.9_{-10.4\%}$ | 6.0 |
| large | | 7.5 | $7.0_{-6.7\%}$ | 6.8 |
| medium.en | 769 M | 8.9 | $8.0_{-10.1\%}$ | 7.1 |
| small.en | 244 M | 12.7 | $10.6_{-16.5\%}$ | 9.0 |
| base.en | 74 M | 32.4 | $17.7_{-45.4\%}$ | 16.1 |

Table 10: WER (%) results of different finetuning methods on CHiME-4 *test-real*. * is the number of trainable parameters. "Full" is full finetuning, "Enc/Dec-only" is encoder/decoder-only finetune.

| Approach | # Param.* | Baseline | STAR | Real |
|---|---|---|---|---|
| *Regular Finetuning* | | | | |
| Full | 1550 M | | $6.0_{-11.8\%}$ | 5.2 |
| Enc-only | 635 M | 6.8 | $6.3_{-7.4\%}$ | 5.0 |
| Dec-only | 907 M | | $6.1_{-10.3\%}$ | 4.4 |
| *Parameter-Efficient Finetuning* | | | | |
| LoRA | 16 M | 6.8 | $6.0_{-11.8\%}$ | 5.1 |
| Reprogram. | 0.4 M | | $6.7_{-1.5\%}$ | 6.7 |

optimal strategy for supervised adaptation. (ii) Using less trainable parameters, LoRA tuning shows commendable results with full finetuning. Nevertheless, LoRA introduces further hyper-parameters (e.g., *rank*) and we found it is sensitive to the learning rate. Considering it slightly decreases the inference efficiency, LoRA tuning is recommended only in situations with limited training resources.

**Analysis of Catastrophic Forgetting.** Following Table 2, we present more results regarding the forgetting issue on Whisper-Medium.en-0.8B and SeamlessM4T-V2-2.3B in Table 11. We observe that our STAR can prevent the catastrophic forgetting problem on different-scale speech foundation models. As for the potential reasons, we analyze from three points. First, we observe that the vanilla self-training scheme can also well mitigate the forgetting problem. We can gain some inspiration from previous work [10] for analysis. Since the pseudo label is generated by the model itself, it may not force the model heavily to over-fit any external data distributions, so that self-training does not degrade the out-of-domain performance. On the other hand, current prevalent foundation models are usually trained by multi-task learning, where not all model capacities are used for ASR. Therefore, specific self-training for ASR task may help mitigate the forgetting problem within ASR though with different domains. Furthermore, compared to vanilla self-training, our STAR shows even better performance. The core contribution of STAR is the novel quality indicator that can better highlight the high-quality pseudo tokens for *informed finetuning*. Considering ASR tasks in different domains, the high-quality pseudo tokens usually correspond to speech frames that are relatively high-quality and easy to recognize. Therefore, highlighting the weights of such pseudo tokens may avoid bringing in low-quality knowledge that may conflict with the existing knowledge embedded in the parameters of the pre-trained speech foundation models, and thus prevent the catastrophic forgetting problem. More study is expected for a deeper understanding of the mechanism behind it. Considering the focus of this work as well as the space limit, we would like to leave this study to future work.

**Iterability of STAR.** As a self-training approach, STAR is iterable by repeating the process of pseudo-labeling and informed finetuning. Table 12 reports the WER results with different numbers of iterations using different model sizes and test sets. In most test sets, multiple iterations of STAR result in further performance improvements. This indicates that while learning from pseudo labels, errors also accumulate, thereby limiting the upper-bound of self-training. Additionally, the enhancement of iteration is relatively larger in smaller models, e.g., 0.7% further WER reduction on Whisper-base.

# F Dataset Domain Details

For dataset selection, our goal is to cover common scenarios of ASR tasks, which can be grouped into three categories, i.e., background noise, speaker accents, and specific scenarios. Consequently, we collect and employ the following datasets with evident domain characteristics to evaluate our proposed approach. In addition, considering our proposed STAR adaptation is built on self-attention matrix that focuses on global contextual correspondence, we filter out short utterances (i.e., with less than 5 tokens) in some datasets for better and more efficient evaluation.

All the data used in this paper are publicly available and under the following licenses: the Creative Commons BY-NC-ND 3.0 License, Creative Commons BY-NC-ND 4.0 License, Creative Commons BY NC-SA 4.0 License, Creative Commons Attribution 4.0 International License, Creative Commons (CC0) License, the LDC User Agreement for Non-Members, the TED Terms of Use, the YouTube's Terms of Service, and the BBC's Terms of Use.

Table 11: WER (%) results regarding catastrophic forgetting with SeamlessM4T-V2 and Whisper-Medium.en as foundation models. "Frozen" denotes zero-shot performance, "Self-train." denotes the self-training baseline. "STAR" denotes that the model is adapted to **CHiME-4** dataset using STAR and then evaluated on other domains. This study is an extension of Table 2.

| Model | LS-FreeSound | | | RATS | CommonVoice | | | | TED-3 | SWBD | ATIS |
|---|---|---|---|---|---|---|---|---|---|---|---|
| | *babble* | *airport* | *car* | | *af* | *au* | *in* | *sg* | | | |
| *SeamlessM4T-V2-2.3B* | | | | | | | | | | | |
| Frozen | 54.0 | 30.1 | **5.1** | 92.7 | **1.8** | **1.7** | **0.8** | **1.1** | 16.3 | 25.9 | 4.3 |
| Self-train. | 52.6 | 30.5 | 5.4 | 92.5 | 2.1 | 2.2 | 1.6 | 1.8 | 15.5 | 26.5 | 4.3 |
| STAR | **44.3** | **28.5** | 5.1 | **88.1** | 1.8 | 1.7 | 1.6 | 1.6 | **10.7** | **21.4** | **3.5** |
| *Whisper-Medium.en-0.8B* | | | | | | | | | | | |
| Frozen | 38.0 | 22.0 | 4.4 | 58.1 | 8.0 | 6.4 | 8.5 | 7.4 | 11.5 | 14.0 | 5.6 |
| Self-train. | 37.8 | 21.0 | 4.5 | 57.7 | 8.2 | 6.3 | 8.6 | 7.4 | 9.8 | 14.4 | 5.9 |
| STAR | **36.1** | **19.8** | **3.8** | **53.4** | **7.8** | **6.0** | **8.0** | **7.1** | **6.4** | **11.6** | **4.3** |
| *Whisper-Small.en-0.2B* | | | | | | | | | | | |
| Frozen | 56.1 | 32.0 | 8.1 | 69.3 | **8.8** | **7.4** | **9.7** | **8.7** | 17.4 | 23.5 | 6.7 |
| Self-train. | 55.3 | 31.0 | 7.2 | 68.3 | 8.9 | 7.8 | 10.1 | 9.2 | 15.4 | 20.2 | 6.1 |
| STAR | **50.7** | **27.4** | **4.6** | **63.0** | 8.9 | 7.7 | 10.1 | 9.6 | **7.0** | **13.8** | **4.5** |
| *Whisper-Base.en-0.07B* | | | | | | | | | | | |
| Frozen | 73.7 | 62.7 | 17.0 | 97.4 | **12.1** | 11.7 | 18.0 | 28.0 | 42.6 | 34.2 | 7.8 |
| Self-train. | 69.6 | 58.8 | 14.5 | 96.8 | 12.5 | 11.4 | 17.8 | 25.4 | 37.7 | 28.8 | 7.0 |
| STAR | **62.0** | **46.8** | **7.9** | **94.0** | 12.6 | **10.9** | **17.4** | **16.5** | **27.2** | **17.6** | **4.9** |

Table 12: WER (%) results of iterative STAR using different sizes of the Whisper ASR models. "# Iterations" denotes the number of iterations of pseudo-labeling and STAR adaptation.

| Model | Test set | # Iterations | | | | | | Real label |
|---|---|---|---|---|---|---|---|---|
| | | 0 | 1 | 2 | 3 | 4 | 5 | |
| large-v3 | *test-real* | 6.8 | 6.0 | 5.9 | 5.7 | 5.7 | 5.7 | 5.2 |
| medium.en | | 8.9 | 8.0 | 7.9 | 7.9 | 7.8 | 7.8 | 7.1 |
| small.en | | 12.7 | 10.6 | 10.3 | 10.3 | 10.3 | 10.3 | 9.0 |
| base.en | | 34.4 | 17.7 | 17.2 | 17.2 | 17.0 | 17.0 | 16.1 |
| | *test-simu* | 9.9 | 9.4 | 9.3 | 9.0 | 8.9 | 8.9 | 8.7 |
| | *dev-real* | 4.6 | 3.9 | 3.9 | 3.8 | 3.8 | 3.8 | 3.2 |
| | *dev-simu* | 7.0 | 6.4 | 6.4 | 6.4 | 6.3 | 6.3 | 5.9 |
| large-v3 | *af* | 6.0 | 4.8 | 4.8 | 4.7 | 4.7 | 4.7 | 4.6 |
| | *au* | 5.8 | 5.1 | 5.0 | 4.6 | 4.5 | 4.5 | 4.3 |
| | *in* | 6.6 | 6.0 | 5.8 | 5.8 | 5.8 | 5.8 | 5.7 |
| | *sg* | 6.5 | 5.1 | 5.1 | 5.1 | 5.1 | 5.1 | 4.9 |

### F.1 Background Noise

**CHiME-4** [83]: CHiME-4 is a popular dataset for far-field noisy speech recognition. It includes real and simulated noisy recordings in four noisy environments, i.e., bus, cafe, pedestrian area, and street junction. We use its *tr05-real* split (9,600 utterances) as the target-domain unlabeled training data, as well as the *test-real* (1,320 utterances), *test-simu* (1,320 utterances), *dev-real* (1,640 utterances) and *dev-simu*(1,640 utterances) splits for testing.

**LibriSpeech-FreeSound** [69]: LibriSpeech-FreeSound is a simulated noisy speech dataset for robust speech recognition, which mixes the clean speech data from LibriSpeech *train-clean-100* split [67] and noise data from FreeSound dataset [22] at SNRs of 0, 5, 10, 15, 20, and 25 dB to simulate the noisy speech data. We randomly select 5,000 long utterances (i.e., with more than 5 tokens) from them as the target-domain unlabeled training data. For test set, they select 118 clean speech samples from LibriSpeech *test-clean* split and mix them with FreeSound noise at SNRs of 0, 5, 10, 15, and 20 dB, where we select three noise types (i.e., babble, airport, car) at 0 dB for main experiments.

**RATS** [26]: The Robust Automatic Transcription of Speech (RATS) dataset contains radio-communication speech in the ultra high-frequency data category that is extremely noisy and challenging for ASR tasks. Its training data contains 43,112 noisy speech utterances, where we filter out the low-quality samples and randomly select 5,000 long samples as training set. Its test set contains 7,591 utterances, where we randomly select 1,000 long samples for higher evaluation efficiency.

### F.2 Speaker Accents

**CommonVoice** [1]: CommonVoice 5.1 is a freely available dataset for speech recognition. It contains speech recordings from diverse speakers in over 60 languages. In this work, we employ the English speech data in four different accents, including African, Australian, Indian, and Singaporean. Specifically, we randomly select 7,885 long samples from its *train-en* split with accent labels, where the training set contains 7,485 samples (1,902/2,000/2,000/1,583 for each accent) and the test set contains 400 samples (100 for each accent). In our experiments, the model is finetuned on the entire training set and then evaluated on each accent individually.

### F.3 Specific Scenarios

**TED-LIUM 3** [29]: TED-LIUM 3 is a dataset of speech recorded from TED Talks in multiple languages. It contains a diverse range of background noise, speaker accents, speech topics, etc. To better evaluate our method, we randomly select 6,000 long samples from its *train* split for our main experiments, where the training set contains 5,000 samples and the test set contains 1,000 samples.

**SwitchBoard** [25]: The SwitchBoard corpus is a telephone speech dataset collected from conversations between pairs of speakers. It focuses on North American English and involves over 2.4k conversations from approximately 200 speakers. We randomly select 5,000 long samples from its *train* split as the training data, and use its *eval2000* split as the test data.

**LRS2** [13]: Lip Reading Sentences 2 (LRS2) is a large-scale publicly available labeled audio-visual dataset, which consists of 224 hours of video clips from BBC programs. We randomly select 6,000 long samples from its *train* split for our main experiments, where the training set contains 5,000 samples and the test set contains 1,000 samples.

**ATIS** [28]: Airline Travel Information System (ATIS) is a dataset comprising spoken queries for air travel information, such as flight times, prices, and availability. It contains 3,964 samples in the training set and 809 samples in the test set, which are recorded from over 500 speakers.

**CORAAL** [43]: The Corpus of Regional African American Language (CORAAL) is the first public corpus of AAL data. It includes audio recordings along with the time-aligned orthographic transcription from over 150 sociolinguistic interviews. We randomly select 2,950 long samples as the training set and 500 samples as the test set.

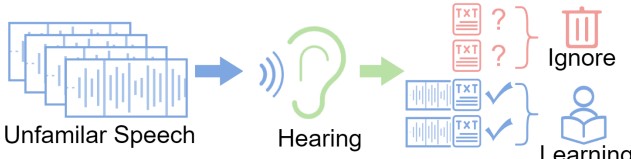

Figure 8: Self-training process of humans. "✓" denotes the self-generated transcription with high subjective confidence that will thus be selected for subsequent learning.

## G  Algorithm of STAR Adaptation

---
**Algorithm 1** Self-Taught Recognizer (STAR) adaptation.
---

**Input:** pre-trained speech foundation model $f^{(s)}$, target-domain unlabeled data $\mathcal{X}^{(t)} = \{x_i^{(t)}\}_{i=1}^{N^{(t)}}$.

**Output:** Target domain ASR model $f^{(t)}$.

Generate pseudo label $\{\hat{y}_i^{(t)}\}_{i=1}^{N^{(t)}}$ from $\mathcal{X}^{(t)}$ using $f^{(s)}$.

**repeat**
    **for** $i = 1$ **to** $N^{(t)}$ **do**
        Collect confidence score $\{\mathcal{C}_l\}_{l=1}^L$ using Eq. (3).
        Calculate attentive score $\{\mathcal{A}_l\}_{l=1}^L$ using Eq. (5).
        Calculate STAR indicator $\mathcal{S}_l$ using Eq. (8).
        Finetune $f^{(s)}$ with $\{x_i^{(t)}, \hat{y}_i^{(t)}\}$ and $\mathcal{S}_l$ using Eq. (8)
    **end for**
**until** ASR model $f$ is converged.

---

