# OpenReview forum: "Self-Taught Recognizer: Toward Unsupervised Adaptation for Speech Foundation Models"
_NeurIPS.cc/2024/Conference — NeurIPS 2024 poster_

### Official Review · Reviewer_8Y8C · 2024-07-08

**Soundness:** 3
**Presentation:** 3
**Contribution:** 3
**Rating:** 6
**Confidence:** 5

**Summary:**

This paper proposed a Self-TAught Recognizer (STAR) that leverages unlabeled data to enhance the robustness of automatic speech recognition (ASR) systems in diverse target domains. The proposed method includes a novel indicator that empirically integrates step wise information during decoding to assess the token-level quality of pseudo labels without ground truth and utterance level pseudo labels filtering . The STAR seems to have sufficient novelty for publication. Sufficient experiments are presented to show the effectiveness of the proposed method. The work should be reproducible since the code will be released.

**Strengths:**

The strengths of the papers are:
1. The proposed method is novel.
2. Sufficient experiments and good analysis.
3. Good presentation and the paper are easy to follow.

**Weaknesses:**

Major concerns about the paper.

1. It was shown in Figure 3 that the UDA converged with about 1 hour speech data for the target domain and the gain is very small with more data. Given that the performance of the STAR is still a lot worse (some are close) than the true-label model (The upper bound of STAR), I am concerning about the usage of STAR in the real scenario. One or several hours data transcription is sometimes affordable for some organizations, so that they might get better numbers than STAR.

2. In other words, UDA has an upper bound while true-label model not. Did authors have the experiments to show how many hours of true-label data finetuning is equivalent to the performance of STAR? This might help the readers understand better on selecting UDA or human-transcription in different scenarios.

**Questions:**

Some minor concerns or suggestions:

1. Maybe I missed the sentence in the paragraph, but I don't find the description of which layer or head of the attention weights are used for the indicator A. It is worth a mention in the paragraph especially in Equation 5.
2. Continuing the indicator A, did the authors compare the confusion matrix for other layers? In other words, does the attentive score confusion matrix behave similarly in Figure 2 for specific layers or for all layers (head)?

**Limitations:**

The limitation of the method can be restricted to certain use cases. However, the proposed method is still helpful to the community.

---

> ### Author Rebuttal · Authors · 2024-08-07
>
> We sincerely appreciate Reviewer 8Y8C for the valuable and constructive comments. Please find detailed responses below:
>
> - ***Weakness 1 & 2***
>
>    Thanks for your feedback. We clarify that the true-label models use all labeled data from the training set of each domain. Therefore, how good this upper-bound is also depends on the size of their training set, with more training details provided in Appendix F.
>
>    Furthermore, fitting a 1B model to a few hours of labeled data can lead to overfitting issues. Without corresponding measures to mitigate this, the model's performance in other domains would significantly degrade. However, the training labels of STAR are self-generated, and reweighting is merely an attempt to selectively update without forcing the model to overfit to a new data distribution. This is also why STAR avoids catastrophic forgetting.
>
> - ***Question 1 & 2: Which layer or head of the attention weights are used for the indicator A.***
>
>    We appreciate your suggestion. In our experiment, the self-attention matrix is drawn from the last transformer layer of the foundation model, averaging on all heads. We also found that the performance is not sensitive to the selection of layers or heads. The last two layers, with a single head, can also achieve comparable performance.
>
>    For the confusion matrix, we confirm that the attentive score from the last five layers is very similar to Figure 2. The earlier layers can also show similar patterns but are somewhat less indicative of the pseudo-label quality, as the high layers learned more linguistic semantics.

---

> > ### Comment · Reviewer_8Y8C · 2024-08-10
> > **Reply the rebuttal**
> >
> > Thank you for the rebuttal. I am keeping my score which is already an accept. Please make sure the explanations can be reflected in the revised paper.

---

> > > ### Author Response · Authors · 2024-08-13
> > > **Thank you for feedback**
> > >
> > > Dear Reviewer 8Y8C,
> > >
> > > Thank you very much for positive feedback! Please do not hesitate to reach us if you have any further questions or comments.
> > >
> > > Best,
> > >
> > > Paper 10759 Authors

---

### Official Review · Reviewer_zXq9 · 2024-07-09

**Soundness:** 3
**Presentation:** 3
**Contribution:** 3
**Rating:** 6
**Confidence:** 5

**Summary:**

The paper proposes STAR, a novel algorithm for unsupervised domain adaptation (UDA) of speech foundation models (e.g., one that has a decoder like Whisper). The UDA setting is the semi-supervised setting where unlabeled data from the target domain is available, and the speech foundation model is available. The unlabeled data is first pseudo-labeled and then used to train the ASR model. The paper specifically proposes the STAR Indicator, which combines the advantages of the auto-regressive confidence score and the attentive score (the attentive score is self-defined by the authors and derived from attention scores) to finally produce a score that can improve the fine-tuning process on pseudo-labels (named as "informed fine-tuning" by the authors and a method prevalent in literature where you guide the training process in a re-weighting manner).

**Strengths:**

The strengths of the paper are as follows:
- The paper is well-written and easy to follow. The illustrations are also well-made.
- The algorithm proposed for uncertainty estimation is novel, to the best of my knowledge.
- The evaluation setup chosen is sufficient, and no further additions can be made to the best of my knowledge.
- I like the pre- and post-result analysis of the paper. Provides intuition into the proposed algorithm and results.
- Section 5.2 and 5.3 is interesting. These kind of analysis is important for speech papers and should be promoted (which is generally not found in other papers).
- The Appendix is well done and provides nice extra details

**Weaknesses:**

- As also claimed by the authors, the overall methodology of using uncertainty estimation for improving UDA is not novel. Only the estimation score proposed is novel. However, these works are properly cited. This makes the paper sound, but not very exciting to me. However, I also acknowledge that this is not a primary reason to reject.
- I am not convinced why only Whisper (and some other models in Table 4) was employed for evaluation (a model that does not have a lot of training details revealed). I have generally found methods evaluated solely on Whisper to not work in a real-world setting. For example, what would have happened if we had adapted a purely LibriSpeech-trained model to the domains mentioned in the paper with the STAR algorithm? Though I understand that the paper title says "Speech Foundation Models," for me, this makes the applicability of STAR a bit less to me. I would leave this decision to other authors
- Some key baselines are missing: What happens to STAR without Utterance-level Filtering? Why was the LM-rescoring baseline not used (LM-rescoring is the most traditional method for UDA where the language model is further trained on target domain texts, may be you can fine-tune the LM on pseudo transcripts? (open to discussion if this is not valid)?

**Questions:**

The algorithm is simple and sound. The evaluation setup is sufficient except the Whisper part. I might have further questions during the rebuttal period after I see other perspectives. But overall, the paper is well done (again, I do not find it too exciting but very sound) and I am inclined towards accepting. I hope the authors can respond to my points in Weaknesses.

**Limitations:**

- I don't see the Limitations of this paper discussed in the main paper. Can the authors please add it to the main paper? Limitations is a very important section of a NeurIPS paper. Additionally, can the authors please elaborate the limitations section with more details like: Latency and resource requirement over plain UDA fine-tuning, high noise scenarios where UDA may fail, etc.

---

> ### Author Rebuttal · Authors · 2024-08-07
>
> We sincerely appreciate Reviewer zXq9 for your valuable and constructive comments. Please find detailed responses below:
>
> - ***Q1: References to previous work***
>
>    We sincerely appreciate your feedback. While this is not a completely new topic, our proposed method only requires a downloadable foundation model and one hour of unlabeled speech to enhance the model in this domain without catastrophic forgetting. With the introduction of more speech foundation models, we believe this is a meaningful exploration.
>
> - ***Q2: What would have happened if we had adapted a purely LibriSpeech-trained model to the domains?***
>
>    Thanks for your question. Allow us to point out that the robustness of a model trained on LibriSpeech is insufficiently robust to serve as a pseudo-labeler. For instance, using the ATIS test set as an example, the LibriSpeech ASR model cannot recognise many city names. In comparison, Whisper, which is a widely used foundation model trained on much larger and more diverse datasets, is a lot more robust and can achieve one-to-many adaptation to accommodate different speech domains. Related discussions are included in Appendix A (Q1 & Q2).
>
> - ***Q3: What happens to STAR without Utterance-level Filtering?***
>
>    Thanks for your question, the ablation study without Utterance-level Filtering is in Table 8, and discussed in Appendix D. Since Table 1 can no longer accommodate additional columns, and Utterance-level Filtering is merely a process of removing particularly difficult-to-recognize samples. The effect is similar across different methods, so we have placed it in the appendix.
>
> - ***Q4: Why was the LM-rescoring baseline not used?***
>
>    Thanks for your suggestion. We need to point out that in the STAR experiments, each domain used only about one hour of unlabeled speech, which is equal to a few hundred utterances (Figure 3). This amount of text is insufficient to directly train or even adapt a useful LM for LM rescoring. Furthermore, all of our experiments were based on a single end-to-end model, while the additional LM-rescoring stage will increase the complexity making it no longer a purely end-to-end system.

---

> > ### Comment · Reviewer_zXq9 · 2024-08-07
> > **Thank You for the rebuttal**
> >
> > Thank You for the rebuttal. I am keeping my score which is already an accept. The paper takes a decent step towards advancing the domain.

---

> > > ### Author Response · Authors · 2024-08-13
> > > **Thank you for feedback**
> > >
> > > Dear Reviewer zXq9,
> > >
> > > Thank you very much for positive feedback! Please do not hesitate to reach us if you have any further questions or comments.
> > >
> > > Best,
> > >
> > > Paper 10759 Authors

---

### Official Review · Reviewer_LswZ · 2024-07-10

**Soundness:** 3
**Presentation:** 3
**Contribution:** 3
**Rating:** 6
**Confidence:** 4

**Summary:**

This paper investigates the use of audio-only data to enhance ASR performance for domain adaption in Speech Foundation Models. The approach is straightforward: recognition results are used to compute a confidence score for each token (e.g., BPE in this paper), which then weights the loss function, as shown in Equation (4). The key innovation is the computation of token-level confidence scores by adopting attention scores instead of the softmax output for confidence estimation. Unlike traditional methods that operate at the utterance level, this approach focuses on token-level confidence. The concept is simple: use the attention weight as the confidence score rather than the predicted next-word probability. The proposed STAR method combines information from both token-level probabilities and the attention matrix. ASR performance appears promising, especially when compared to the strong baseline model Whisper large v3.

**Strengths:**

1. I find the proposed confidence estimation using attention scores intriguing.

2. The authors conducted extensive experiments on various ASR corpora to demonstrate the effectiveness of the proposed STAR algorithms, along with comprehensive ablation studies.

3. Significant performance improvements are reported over the strong baseline, which is Whisper large v3.

**Weaknesses:**

1. Several details are missing. How is the attention matrix calculated given multiple layers and heads? Are the attention weights from cross-attention or self-attention used?
2. Do you need to tune the threshold for each corpus, or simply use a fixed threshold for all datasets? What is the specific value for the threshold lambda?
3. The definition of conflict is A_l^2/C_l, which seems intuitive. Is there any particular reason for using this formulation?
4. Many studies have attempted to use entropy from the output layer instead of probability for confidence estimation in end-to-end ASR systems.
5. I am interested in whether the informed fine-tuning shown in Equation (4) is proposed in this paper. The confidence score is used to filter utterances and weigh the importance of each word. Do you have any ablation studies demonstrating the effectiveness of this approach?
6. It would be better to combine Equations (6) and (7) into an equation array to clearly state the formulation.
7. In Equation (5), the attention score is computed using the attention weight for the current word to the previous words and from future words to the current word. Is there any particular reason for this choice? Is it acceptable for the A_l to be larger than 1?

Overall, this paper seems to rely heavily on empirical or intuitive settings, making it difficult to extend to a theoretically sound approach. Additionally, many key details are missing. It would be appreciated if the authors could provide more details and present the information more clearly.

I am open to reconsidering my rating based on the authors' rebuttal.

**Questions:**

I have listed my questions in the above section.

---

> ### Author Rebuttal · Authors · 2024-08-07
>
> We appreciate Reviewer LswZ for valuable and constructive comments, and we believe our detailed responses below can solve your concerns on missing details. Please let us know if you have further questions or recommendations.
>
> - ***Q1: Several missing details: Are the attention weights from cross-attention or self-attention used? How is the attention matrix calculated given multiple layers and heads?***
>
>    Thanks for your question. We use self-attention weight as we introduced in the paper (**line 65** and **line 180**).
>
>    In our experiments, the attention matrix is obtained by averaging all heads in the last transformer layer, as the high layers learned more linguistic semantics. We also found that the performance is not sensitive to the selection of layers or heads. The last two layers, with a single head, can also achieve comparable performance.  More details can be found in our submitted code.
>
> - ***Q2: Do you need to tune the threshold for each corpus, or simply use a fixed threshold for all datasets? What is the specific value for the threshold lambda?***
>
>    Thanks for your question. We apply a fixed threshold of lambda$=2$ for all corpus and it works well generally, as mentioned in **line 264**.
>
> - ***Q3: The definition of conflict is $A_l^2/C_l$, which seems intuitive. Is there any particular reason for using this formulation?***
>
>    Thanks for your question. The design of $A_l^2/C_l$ can be decoupled into two terms, $A_l/C_l$ and $A_l$, which means to match the “conflict” criterion, we not only require $A_l$ to be much larger than $C_l$, but the absolute value of $A_l$ should also be large. This is to avoid special cases with two small scores where one is many times the other (e.g., $A_l = 0.1, 0.01$). We have mentioned this particular design in **line 212 to 214** and **footnote 2**.
>
>    The specific equation definition of “conflict” is not unique, if only the above condition is met roughly.
>
> - ***Q4: Many studies have attempted to use entropy from the output layer instead of probability for confidence estimation in end-to-end ASR systems.***
>
>    Thanks for your question. Entropy is indeed an alternative choice for confidence estimation. However, in our case, since the probability $P$ cannot reliably indicate pseudo-label quality (Figure 2, Figure 5, and Figure 6), the minus entropy $P*logP$ that is positively correlated to $P$ cannot serve as a reliable indicator either. Our preliminary results have confirmed this, but considering $P$ is a more typical and intuitive choice, we only select $P$ as the confidence score baseline.
>
> - ***Q5: I am interested in whether the informed fine-tuning shown in Equation (4) is proposed in this paper. The confidence score is used to filter utterances and weigh the importance of each word. Do you have any ablation studies demonstrating the effectiveness of this approach?***
>
>    Thanks for your question. The informed finetuning with confidence score in Eq. (4) is followed from prior works as discussed in the Introduction (**line 59 to 62**). However, we observe that the confidence score is unreliable in assessing pseudo labels so we propose a reliable attentive score based on the self-attention matrix. Finally, we combine conventional confidence and proposed attentive score as our final STAR indicator, which guides the token-level reweighting. On the other hand, utterance-level filtering is implemented by Monte-Carlo Sampling introduced in Section 3.3.
>
>    For experiments, Table 1 presents the ablation study of utterance-level filtering, and token-level reweighting (confidence score, attentive score, STAR score).
>
> - ***Q6: It would be better to combine Equations (6) and (7) into an equation array to clearly state the formulation.***
>
>    Thanks for your suggestion and careful check, we will revise it in the next version.
>
> - ***Q7: In Equation (5), the attention score is computed using the attention weight for the current word to the previous words and from future words to the current word. Is there any particular reason for this choice? Is it acceptable for the A_l to be larger than 1?***
>
>    Thanks for asking this question. For the choice of history and future tokens in calculating the attentive score (Eq. (5)), we use both of them to capture the comprehensive global context to better assess the role of the current token, in terms of its semantic significance in the entire sentence. Table 7 presents an ablation study on this choice, indicating that both history and future tokens are helpful. The $A_l$ will be normalized using sequence length after calculation so that the absolute value of $A_l$ does not matter, we will add these details in the next version.

---

> ### Author Response · Authors · 2024-08-13
> **Regarding the deadline of author-reviewer discussion period**
>
> Dear Reviewer LswZ,
>
> Thank you for your kind efforts in providing initial reviews for our work! We have taken them into careful consideration and provided the responses accordingly.
>
> Since the deadline of author-reviewer discussion period is approaching, could you please take some time to confirm whether our responses have satisfactorily addressed your concerns? If you believe so, may you please consider adjusting your initial score accordingly? Please do not hesitate to reach us if you have any further questions or comments.
>
> Best,
>
> Paper 10759 Authors

---

> > ### Comment · Reviewer_LswZ · 2024-08-14
> > **Thank You for the rebuttal**
> >
> > I appreciate the authors' detailed rebuttal addressing the concerns I highlighted in my review. I think the rebuttal addresses some of my questions effectively. I have adjusted my score accordingly.

---

### Official Review · Reviewer_vnSx · 2024-07-10

**Soundness:** 3
**Presentation:** 4
**Contribution:** 3
**Rating:** 7
**Confidence:** 4

**Summary:**

The paper proposes STAR, a novel ASR domain adaptation technique that requires no labeled data and only a few unlabeled samples. STAR utilizes the confidence score and self-attention score obtained during decoder inference to calculate the reliability score (STAR indicator) for each token. The score of each token is then used as a multiplier to adjust the fine-tuning loss, which is based on generated pseudo-labels. In addition, STAR employs utterance-level filtering to remove noisy predictions. Extensive experiments across various ASR models, datasets, and fine-tuning techniques demonstrate that STAR achieves significant accuracy improvements compared to the baseline self-training approach.

**Strengths:**

* The logical flow of this paper is very interesting, and the authors provide empirical evidence for each step. The motivations behind the research and method are inspiring.
* The authors have conducted comprehensive experiments using multiple datasets, models, and noisy conditions. I appreciate the authors’ effort on this.
* STAR does not seem to suffer from catastrophic forgetting, and this is a very important advantage.

**Weaknesses:**

* The proposed method is designed for a Transformer-based model with auto-regressive generative decoder architectures. As such, it may not be easy to adopt STAR for CTC or RNN-T-based ASR models (as the authors also discussed in Appendix A).
* It would be important to discuss the differences and similarities between STAR and noisy student-based training [1][2]. NST also employs pseudo-labeling, heavy filtering, and iterative refinement.
  * [1] Pushing the Limits of Semi-Supervised Learning for Automatic Speech Recognition
  * [2] Improved Noisy Student Training for Automatic Speech Recognition

**Questions:**

* The attentive score A (Equation 5) seems to be affected by the total length of the generated transcript. The longer the transcript, the (potentially) higher the attentive score. In contrast, confidence scores are bounded to [0, 1]. Maybe missing a normalization term in Equation 5?
* Any intuitive reasons not to incorporate cross-attention scores?
* Comparing Table 1 and 2, the numbers of “Frozen” are the same but “Self-train” and “STAR” are different. For example, TED3 WER is (5.2, 4.9, 4.1) in Table 1 but (5.2, 5.3, 5.0) in Table 2. What’s the difference?
* How many beams are used in the beam search? Are the beam search-based pseudo-labels also used for self-training baselines?
* It would strengthen the paper’s claim if the pseudo-label and STAR score could be cross-transferred between different models (for example, Whisper-Large generates training resources for Whisper-Tiny).

**Limitations:**

* The paper adequately addresses the limitations.

---

> ### Author Rebuttal · Authors · 2024-08-07
>
> We sincerely appreciate Reviewer vnSx for valuable and constructive comments. Your suggestion is also instructive for further analysis and our future work.
>
> Please find detailed responses below:
>
>
> - ***Q1: Appling STAR for CTC or RNN-T-based ASR models.***
>
>    Thanks for your feedback, As the illustration in section 3, applying STAR to CTC or RNN-T, auto-regressive Whisper is still employed as a pseudo-labeler to provide pseudo transcription, as well as token-level or utterance-level indicators. Therefore, the RNN-T model can be adapted based on the provided information. We acknowledge that the process requires compositional efforts, but this connection to popular ASR variants also has high potential application value.
>
> - ***Q2: Discussion of the differences and similarities between STAR and noisy student-based training [1][2].***
>
>    Thanks for your feedback. We will add a paragraph of related discussion to the paper. Although both STAR and NST contain a pseudo labeling and filtering process, they have several distinctions listed as follows:
>
>    - **Motivation:** STAR focuses on domain adaptation, and NST focuses on leveraging abundant unlabeled data to improve general ASR accuracy, where the unlabelled data are from similar domains (LibriSpeech and LibriLight datasets).
>    - **Data:** STAR only requires **1 hour** of unlabeled data from any target domain, NST requires both labeled data (960 hours) and abundant unlabeled data (60k hours).
>    - **Method:** STAR focuses on the token-level indicators for informed finetuning (which could be attributed to the strong linguistic ability of the Whisper decoder), and NST focuses on utterance-level data filtering.
>
> - ***Q3: Missing a normalization term in Equation 5.***
>
>    Thanks for your valuable question. When we obtain the attention matrix, it is normalized according to the sequence length. We will clarify  this point with a note.
>
> - ***Q4: Why not use the cross-attention matrix?***
>
>    Thank you for this discussion. As we described in the paper, self-attention in the decoder better reflects the **linguistic acceptability** of the predicted transcription. As for cross-attention, it is more directly linked to the acoustic representation–vulnerable to variable speech domain. In contrast, self-attention is text-based and Whisper's multi-task pre-training also includes text-only learning objectives. Therefore, self-attention potentially gives more linguistic knowledge. Our preliminary results also confirmed that the cross-attention matrix could not provide a reliable indicator.
>
> - ***Q5: Difference between Table 1 and Table 2.***
>
>    Thanks for your question. Table 1 shows the main results where Whisper is both finetuned and tested on each domain individually. Table 2 aims to explore the forgetting issue of STAR, where Whisper is finetuned on the CHiME-4 dataset and then tested on other datasets. Therefore, Table 2 presents worse results than Table 1. We will clarify this in the captions of the tables.
>
> - ***Q6: How many beams are used in the beam search? Are the beam search-based pseudo-labels also used for self-training baselines?***
>
>    Thanks for your question. For beam search, we set beam$=5$. Our preliminary results show that beam search-based pseudo labels show almost the same self-training performance as the greedy search ones. Therefore, for higher efficiency, we use greedy search only.
>
> - ***Q7: ​​Can Whisper-Large generate training resources for Whisper-Tiny?***
>
>    Thanks for the question. Yes, we confirm that STAR can distill knowledge from large models to smaller models (but not vice versa). We will include this advantage in the next version.

---

> > ### Comment · Reviewer_vnSx · 2024-08-08
> > **Thank you for the response**
> >
> > Thank you for addressing my questions. I am keeping my score toward accept, and I believe this paper can inspire many following studies.

---

> > > ### Author Response · Authors · 2024-08-13
> > > **Thank you for feedback**
> > >
> > > Dear Reviewer vnSx,
> > >
> > > Thank you very much for positive feedback! Please do not hesitate to reach us if you have any further questions or comments.
> > >
> > > Best,
> > >
> > > Paper 10759 Authors

---

### Decision · Program_Chairs · 2024-09-25

**Decision:**

Accept (poster)

**Comment:**

This paper is one of those just above the acceptance threshold. The questions from the reviewers are fair. In particular, the novelty is somewhat lacking (questioned by vnSx and zXq9). Given that the approach is simple and the improvement is sizable, I am recommending an acceptance for this one.

I recommend the authors revise the parts of the paper that causes confusion, in particular, the parts that reviewers raise questions. Though the confusion was later resolved in the discussion, that means the paper requires revision.